# Diagnosis, Treatment, and Management of Otitis Media with Artificial Intelligence

**DOI:** 10.3390/diagnostics13132309

**Published:** 2023-07-07

**Authors:** Xin Ding, Yu Huang, Xu Tian, Yang Zhao, Guodong Feng, Zhiqiang Gao

**Affiliations:** Department of Otorhinolaryngology Head and Neck Surgery, The Peaking Union Medical College Hospital, No. 1, Shuaifuyuan, Dongcheng District, Beijing 100010, China; darcy980105@163.com (X.D.); hy1345362083@163.com (Y.H.); tianxu214@163.com (X.T.); hanspro@126.com (Y.Z.)

**Keywords:** artificial intelligence, machine learning, deep learning, otitis media

## Abstract

A common infectious disease, otitis media (OM) has a low rate of early diagnosis, which significantly increases the difficulty of treating the disease and the likelihood of serious complications developing including hearing loss, speech impairment, and even intracranial infection. Several areas of healthcare have shown great promise in the application of artificial intelligence (AI) systems, such as the accurate detection of diseases, the automated interpretation of images, and the prediction of patient outcomes. Several articles have reported some machine learning (ML) algorithms such as ResNet, InceptionV3 and Unet, were applied to the diagnosis of OM successfully. The use of these techniques in the OM is still in its infancy, but their potential is enormous. We present in this review important concepts related to ML and AI, describe how these technologies are currently being applied to diagnosing, treating, and managing OM, and discuss the challenges associated with developing AI-assisted OM technologies in the future.

## 1. Introduction

Otitis media (OM) is one of the most common pediatric diseases, affecting almost two-thirds of preschoolers [1]. Unfortunately, doctors often fail to correctly diagnose OM [2,3], partly due to the young age of patients, poor awareness of patients and their families, and the insidious nature of the symptoms of the disease. The absence of an early diagnosis of OM can lead to hearing loss, speech impairment, and even serious extracranial and intracranial complications [4,5]. Due to many misdiagnosed cases of OM, antibiotics are overused and unnecessary operations of OM are performed, which also places a burden on global public health [6,7]. While at the same time, the advent of big data and artificial intelligence (AI), as well as the wide use of electronic health records, present an opportunity to revolutionise how patients with OM are treated. Based on a large amount of data collected over time about OM patients, AI can be used to identify latent diseases, narrow down disease types, predict disease risks, personalise treatment options, assist in surgery, and optimise disease management (Figure 1) [8]. This article discusses AI and machine learning (ML) as contributions to OM management, the need for AI algorithms tailored to OM, and the implications of integrating AI and OM care for patients, physicians, surgeons, and the entire multidisciplinary team of care providers.

## 2. Artificial Intelligence & Machine Learning

### 2.1. What

In a broad sense, AI is the ability of machines to mimic the functions of human intelligence. As a form of AI, ML uses statistical methods to enable machines to learn tasks without the need for explicit programming [9]. Often, learning involves mathematical model fitting. Mathematically, relationships between features (or variables) are analysed through many data points, resulting in an optimal fitting model. Deep learning (DL) extends ML, while mathematical models are more complex and detailed.

### 2.2. How

ML relies on data and algorithms, and a large amount of data and advanced algorithms, if trained and adjusted appropriately, can often produce ideal results (Table 1 and Figure 2). Clinical data regarding OM patients can be collected more easily with the widespread use of electronic health record systems. Data included medical history records, physical examination records, laboratory results, otoscopy images, audiology room images, imaging data, drug use records, surgical records, surgical videos, and pathological records. A large amount of data is being collected in the healthcare field from large-scale genetic research and consumer electronic devices (such as wearables and smartphones) that provide physiological and behavioral information.

### 2.3. Why

By contrast with traditional statistical methods based on hypothesis, ML algorithms can be fitted with large amounts of data to determine the model that will produce the best results. In addition to linear relationships between features and results, ML algorithms can capture more complex, nonlinear interactions [10,11]. ML can also handle large data sets and integrate insights from different data types to provide precision medicine solutions.

## 3. Materials and Methods

The present review was conducted in accordance with the PRISMA guidelines. The overall workflow is shown in Figure 3 and described below. Ethics approval and patient consent were not required for this review.

### 3.1. Literature Search

A search of online databases (including PubMed, MEDLINE, EMBASE, Scopus and ResearchGate databases) for articles, abstracts or conference proceedings published between 2011 and 2022 that used Al-based approaches to diagnose, treat or manage OM patients was conducted. Searches were limited to those involving human subjects and those published in the English Language.

Medical subject headings (MeSH) terms and non-MeSH terms related to Al approaches included: ‘artificial intelligence’, ‘machine learning’, ‘deep learning’, ‘convolutional neural networks’, ‘data mining’, ‘computer-assisted diagnosis’, ‘computer-assisted surgery’, ‘natural language process’ and ‘computer vision’. MeSH terms and key-words related to otitis media included: ‘ear’, ‘eardrum’, ‘tympanic membrane’, ‘ear disease’, ‘otitis’, ‘otitis media’, ‘acute otitis media’, ‘chronic otitis media’, ‘chronic suppurative otitis media’, ‘cholesteatoma’, ‘OM’, ‘AOM’, ‘OME’, ‘COM’, ‘CSOM’.

### 3.2. Selection Criteria

Titles and abstracts were reviewed for eligibility by two independent investigators (Ding, X. and Huang, Y.). Discrepancies between the two investigators were resolved by two senior authors (Feng, G. and Gao, Z.), board certified otolaryngologists. Reference lists of available full-text articles were also manually screened for further studies eligible for inclusion in this review. Studies describing the development of an autonomous algorithm to diagnose, treat or manage OM patients using the Al approaches described above were included. Articles that were excluded consisted of those that published in the non-English Language, did not use Al-based approaches, explore other diseases instead of OM or were review articles or editorials. Study inclusion/exclusion is summarised in a PRISMA flow diagram (Figure 3).

### 3.3. Data Extraction

Two investigators (Ding, X. and Huang, Y.) independently extracted data from included studies for analysis. The following characteristics were extracted and listed from included studies: primary author, year, study objective, Al technique to achieve study objective, number and datasets, deep learning models and performance (Appendix A).

## 4. Diagnosis

Below, we will proceed to describe in detail recent progress in the diagnosis of otitis media based on AI. This section was divided into *Computer Vision* (Section 4.1) and *Natural Language Processing* (Section 4.2) according to the CNN methods, and the Computer Vision part was further classified into Otoscopy (Section 4.1.1), Radiology & Pathology (Section 4.1.2) and Tympanometry (Section 4.1.3) according to the data sources.

### 4.1. Computer Vision

#### 4.1.1. Otoscopy

A branch of AI known as computer vision (CV) uses deep learning algorithms to analyse images and videos automatically. Recently, CV has shown great promise in OM-related medical images, particularly otoscopy images. Three main types of ear endoscopes are commonly used in clinical practice: the endoscopic otoscope, the electronic otoscope, and the video pneumatic otoscope (VPO). In recent years, special otoscopes have been developed, such as multicolor imaging otoscopes, shortwave infrared otoscopes, and optical coherence tomography (OCT) in vivo. After screening relevant articles on the analysis of OM endoscopic images based on AI, a total of 43 relevant articles were identified (Appendix A) [12,13,14,15,16,17,18,19,20,21,22,23,24,25,26,27,28,29,30,31,32,33,34,35,36,37,38,39,40,41,42,43,44,45,46,47,48,49,50,51,52,53,54]. In most studies, the ML system achieved a high level of diagnosis and was as accurate as the clinician or even better. A new ML algorithm has been developed by Matthew G. Crowson et al., using 639 otoscopic images containing middle ear effusions and infections. On a test set consisted of 22 images, their algorithm detected abnormalities in the tympanic membrane with an accuracy of 95.5%, significantly higher than the 65% average accuracy of 39 pediatric and ENT physicians [15]. Moreover, Hayoung Byun et al., further constructed a convolutional neural network (CNN) to identify conductive deafness based on 1130 VPO images of patients with OM. With an accuracy of 94.1% (AUC 0.972), their CNN algorithm could distinguish between normal tympanic membranes, conductive hearing loss and sensorineural hearing loss, which is higher than the accuracy rate of 79.0% (AUC 0.773) of experienced ENT physicians [20]

At the same time, many research teams are also focusing on developing special ear endoscopy technology based on AI to diagnose OM. Using the single green channel mode, for example, Michelle Viscaino et al., demonstrated the optimal overall diagnostic performance (accuracy 92%, sensitivity 85%, specificity 95% and precision 86%) by changing the otoscope colour based on multicolour imaging [21]. Rustin G. Kashani et al., developed an algorithm that could accurately detect effusion when OM was present, using 1179 shortwave infrared otoscopic images [29]. Optical coherence tomography (OCT) is a non-invasive test that evaluates the tympanic membrane (TM) and the adjacent middle ear space (OM). Using OCT images, otoscopic images and patient reports, Guillermo L. Monroy et al., trained a neural network capable of automatically identifying the TM and middle ear effusion in OM patients with a diagnostic accuracy of more than 90% [55]. They further increased the OCT images of the chinchilla, the gold-standard pre-clinical animal model used to study human OM in subsequent studies to increase the sample size and robustness, further improving the accuracy of their model (95% accuracy) [56].

#### 4.1.2. Radiology & Pathology

An optimal diagnostic imaging method for chronic otitis media (COM) is high-resolution computed tomography (CT) of the temporal bone since it can provide detailed information about the middle ear structure and shows great sensitivity in detecting the extent and complications of middle ear lesions. Due to the rapid development of deep learning technology, computer vision processing and recognition techniques have been applied to interpret CT images of OM automatically (Appendix A) [57,58,59,60,61]. Using this method can greatly reduce labor costs and human errors due to repetition, fatigue, or knowledge differences, as well as improve the accuracy of the diagnosis. Yanmei Wang et al., developed an algorithm for diagnosing COM by analysing 1147 CT images containing cholesteatoma and chronic suppurative otitis media (CSOM). Their model reached an AUC of 0.92, with sensitivity and specificity exceeding the average for clinical specialists (sensitivity 83.3% vs. 81.1%, specificity 91.4% vs. 88.8%) [57]. Using the ROI method of Mask R-CNN, Zhenchang Wang et al., conducted the image extraction network. Their model further improved the accuracy of the pathological recognition algorithm (AUC 0.96) compared to Yanmei Wang et al. [58]. Bo Duan et al., used GoogleNet to develop a neural network capable of distinguishing primary ciliated dyskinesia (PCD)-associated OM from otitis media effusion (OME). Their model exhibited a diagnostic accuracy of greater than 90% [61].

Even though CT has many advantages in diagnosing OM, plain radiography or magnetic resonance imaging (MRI) are still used clinically in certain cases. Considering the sensitivity to radiation, plain radiographs were selected for auxiliary diagnosis of OM in children under two years of age. A neural network was trained to diagnose mastoiditis based on 9988 images of mastoid X-rays obtained from 4994 patients by Kyong Joon Lee et al. Their model demonstrated a high diagnostic accuracy (AUC 0.968 ± 0.003) [62]. In addition to being radiation-free, MRI can distinguish cholesteatoma of the middle ear from cholesterol granuloma, exhibiting high signals on both T1 and T2 sequences. In order to diagnose diseases more quickly and accurately, the combination of pathological sections and AI is considered the gold standard. Although AI-assisted MRI and pathological diagnosis have been extensively studied in cardiovascular, tumor and other diseases, research on OM remains limited.

Radiomics is a rapidly growing field of research involving the extraction of quantitative indicators in medical images, known as radiomic features. Radiomic features capture tissue and lesion characteristics, such as heterogeneity and shape, that can be used alone or in combination with demographic, histological, genomic or proteomic data to solve clinical problems. The approach relates the omics with radiology forming a bridge between two areas of research that may prove useful for clinical treatment planning leading to better outcomes. Although radiomics has shown great effects in the diagnosis, treatment and prognosis of various diseases, especially tumor, there are few relevant studies in the field of OM [63,64].

#### 4.1.3. Tympanometry

Tympanometry measures the sound quality of the ear canal by using an acoustic probe (usually 226 Hz or 1 kHz) and a microphone (for measuring sound). In addition, it can provide us with quantitative information regarding the TM and tympanum of the middle ear, as well as determine whether or not there is any fluid present(Appendix A) [41,65,66,67]. A decision fusion method based on majority voting was adopted by Hamidullah Binol et al., to analyse 73 records with both tympanometry and digital otoscopy data, and their model classification accuracy reached 84.9% [41]. The wideband tympanometry (WBT) method is an extension of standard tympanometry that simultaneously determines the full range of middle ear characteristics at the audiometrically most important frequencies, thereby achieving a more accurate and efficient assessment than a standard 226 Hz or 1 kHz tympanogram. The work by Josefine Vilsøll Sundgaard et al., applied deep learning to WBT, using 1014 measurements for training, and the algorithm reached an average accuracy of 92.6% for the detection of OMs. Nevertheless, the model does not distinguish well between specific types of OM, such as acute otitis media (AOM) and OME [65].

### 4.2. Natural Language Processing

A large amount of textual data in electronic medical records and laboratory tests allow natural language processing (NLP) to automatically identify characteristic descriptive words, thereby detecting cases of OM (Appendix A) [68,69,70,71]. Kuruvilla A et al., applied rule-based natural language processing algorithms to the descriptive words of patients’ tympanic membranes to detect patients with OM automatically. First, the authors identified 8 characteristic words related to the tympanic membrane of OM (such as tympanic membrane bulging, bubbles presence, central concavity, translucency, etc.) based on the analysis of 783 tympanic membrane image examination reports of children from otoscopic experts. Following that, they used the random forest classification classifier algorithm to establish the appropriate syntax and finally developed the NLP model for the diagnosis and classification of OM. The algorithm they developed achieved an accuracy level of 89.9%, which is significantly higher than clinicians without special training [68]. H Binol et al., also used natural language processing and the bag-of-words model (BOW) to analyse otoscopic reports of OM patients to determine five characteristic vocabulary words that could be used to diagnose OM. Their proposed model provided an overall F1-score of 90.2% based on neighborhood component analysis (NCA) [69]. In addition to constructing the NLP model using note texts and structured data from children under five, Herigon JC et al., also limited the training object to otoscopic report findings. Their model can be used to identify patients with AOM more accurately to assess antibiotic prescribing practices better [70].

## 5. Treatment

### 5.1. Internal Medicine

Currently, the primary treatment for OM is the use of antibacterial drugs. However, paediatricians and ENT doctors are constantly concerned about the long-term safety of drugs and the emergence of antibiotic resistance. The use of ML models can allow us to predict individual susceptibility to antibiotics and resistance to them, improve the effectiveness of antibiotics, and even design new antibacterial drugs. In general, polymyxins are used as a last-resort treatment for Gram-negative infections, and Klebsiella pneumoniae is one of the most common Gram-negative bacteria resistant to polymyxins. According to Nenad Macesic et al., ML was applied to whole-genome sequencing data from >600 Klebsiella pneumoniae clonal group 258 (CG258) genomes, and this approach gave an accurate prediction of polymyxin sensitivity and resistance (AUC 0.894) [72]. A study by Rachael A. Mansbach et al., combined ML with drug development, analysed the representation and definition of compounds using an ML algorithm, determined the characteristic submolecular fragment vocabulary, and fitted the prediction model according to the vocabulary. A set of fragments identified by their algorithm can allow drugs to penetrate the outer membrane of Pseudomonas aeruginosa bacteria, in addition to nine compounds that are regarded as good outer membrane penetrators, five of which have been successfully validated experimentally [73].

Increasing resistance to traditional antibiotics has led many researchers to focus on developing new antibacterial drugs. AI can greatly reduce drug development’s time and capital costs when used as an intelligent auxiliary tool. Francisco R Fields et al., used ML and minimum-region determination methods based on biophysics to automate the analysis of antimicrobial peptides, a type of antibiofilm drug, for identifying antimicrobial peptides suitable for synthesis and therapeutic use. In their model, 28,895 20-mer candidate peptides were evaluated for their biophysical properties. Significant antibacterial activity was observed against *Escherichia coli* and *Pseudomonas aeruginosa* for the peptides with the highest overall score, indicating that the method is useful for discovering and synthesising novel candidate bactericin [74]. Deepmind developed AlphaFold in 2021. Using this computer algorithm, it is possible to predict the three-dimensional structure of a protein based solely on its amino acid sequence and to demonstrate its accuracy through further experiments [75]. AlphaFold2 was updated again later by the Deepmind team. In a recent study, David Baker et al., devised RoseTTAFold, a “three-track” neural network capable of analysing the three-dimensional structure of a sequence of proteins within 20 min [76]. The prediction of drug action sites and target protein structures is critical to designing new drugs, predicting drug-receptor interactions, and improving understanding of drug efficacy, adverse reactions, and multidrug interactions.

Nanomaterials loaded with drugs can more accurately reach the lesion area and achieve better therapeutic effect. Fluorescent nanodiamonds (FNDs), possessing extraordinary optical and electronic properties, were used for the ultrasensitive disease and virus detection and the drug delivery. Combined with AI, fluorescence-based biosensors can achieve more accurate and real-time detection and monitor under inevitable conditions or contamination [77]. This may provide new ideas for the rapid diagnosis and targeted drug therapy of OM.

### 5.2. Surgery

As the middle ear has a small surgical space and delicate components, it is important to identify significant landmarks to perform the procedure accurately and safely. Several teams have been working in OM-related surgery to conduct corresponding research and make certain advances (Appendix A) [37,78,79,80,81,82,83,84,85,86,87,88,89,90,91,92]. For example, Toru Miwa et al., developed an AI system to detect cholesteatoma lesions during otoscopic surgery. It should be noted that, even though the detection accuracy of their algorithm was very limited, it was superior to non-otologists in identifying the cholesteatoma matrix (42.3% vs. 38.5%) [80]. Using a neural network, Andy S. Ding et al., were able to segment anatomical structures in the temporal bone with a submillimeter level of accuracy. As a result, this algorithm achieved a high detection accuracy (six of sixteen significant landmarks, Dice 0.681 ± 0.157) for auditory ossicles, including malleus and incus, important nerves, including cochlear nerve and facial nerve, and important blood vessels, including internal carotid artery and sigmoid sinus [81]. This technology can enhance existing image guidance technologies by informing surgeons of potential contact with critical anatomy in real-time. Using the videos of patients undergoing OM surgery under the microscope, our team developed a neural network capable of identifying the chorda tympani (CT) and the short process of the incus (SPI). The recognition accuracy of our model is fairly high. In addition to locating the facial nerve and other important anatomical structures, subsequent registration technology helps to lay the foundation for surgical navigation.

## 6. Risk Prediction & Postoperative Care

Although many epidemiological studies have been conducted on OM as a common respiratory complication in children, few studies have been able to analyse big data so that symptom severity, longitudinal patterns and variance in OM patients related to age and seasonality can be accurately described. In order to plan primary health care services and individualised treatments effectively, these factors are important (Appendix A) [93,94,95]. NLP was used by Anthony Dowell et al., to assess primary care incidence and service utilisation of childhood respiratory illnesses in 77,582 children. Additionally, they demonstrate that the same approach to AI-assisted analysis of big data in healthcare can be applied to other health conditions [93].

COM is typically treated with tympanoplasty. This procedure is intended to improve hearing and repair perforated eardrums. The study by Joanna Szaleniec et al., developed a model that could predict whether a patient’s hearing improved after tympanoplasty utilising feedforward artificial neural networks (FNN) and 21 independent variables (such as age, gender, ear pathology, surgical procedure description, etc.). Their algorithm was able to make 84% of correct predictions [94]. Hajime Koyama et al., developed a model to predict the air-bone gap (ABG) after tympanoplasty using random forest. They found that the correct prediction percentages of their algorithm for predicting postoperative ABG ≤ 15 or >15 dB were 81.5%. They also determined the factors, such as age and ABG, influencing the outcome of tympanoplasty [95]. ENT specialists have also become interested in the study of intelligent follow-up regarding the presence and patency of tympanostomy tubes following their placement. Xin Wang et al., used a support vector machine (SVM) to construct a neural network capable of detecting the presence of a tympanostomy tube in an otoscopic image. Notably, their model achieved an accuracy of 90%, which can be very useful for regularly scheduled follow-ups to check the status of the tympanostomy tubes [79].

## 7. Challenges and Future Considerations

### 7.1. Data & Algorithm

It is generally recognised that deep learning requires large data sets to improve its robustness. However, most of the studies are still single-center and small-sample studies. In particular, very few studies have used OM sets in NLP. In addition, many studies intentionally exclude atypical data when establishing their data sets. For example, many otoscopic images that are difficult to identify due to bleeding, blurring, or defocus are often excluded from studies related to otoscopy. While this improves the accuracy of its reporting in the paper, it significantly reduces its robustness. In terms of algorithms, we still lack high-precision and lightweight algorithms, especially for research requiring real-time tracking and recognition, such as surgery, where recognition speed plays an important role in the practical application of AI.

### 7.2. Application

Various measures are often used to diagnose and treat certain diseases in real clinical practice. An accurate diagnosis of OM requires the combination of the patient’s medical history, otoscopic images, audiogram, CT scan, and other inspections for a comprehensive analysis. Despite this, most researchers tend to focus on a specific method for using AI to assist in diagnosing and treating OM patients. A comprehensive AI-assisted diagnosis, treatment, and disease management system must be developed in the future that combines multiple measures to enhance the potential of AI-assisted medicine. Additionally, it should be noted that most current AI-assisted OM studies are based on retrospective data, but the real world is filled with uncertainties. Future relevant studies should demonstrate the feasibility of AI in practice by applying it to real-world clinical scenarios. Lastly, when AI is applied to solve relevant clinical problems, the “black box” nature of the deep learning network model makes it difficult for users to understand its decision-making mechanism, which not only impedes the optimisation of model structures and safety enhancement but also greatly increases the cost of training and parameter adjustment. While the Grad-CAM can now be used to explain the regions of interest and the decision principles in some network models, the current Explainable Artificial Intelligence (XAI) methods still have significant limitations [96]. We should continue to develop appropriate methods for interpreting deep learning models vigorously in the future so that clinicians may better understand and use relevant AI tools and provide a scientific basis for optimising the structure of deep learning models.

### 7.3. Privacy & Regulation

Patient privacy is an important consideration in developing AI and other information technologies. However, sharing information is an inevitable trend in establishing multi-center and big datasets. Consequently, ensuring the privacy and confidentiality of patient information is an important issue that needs to be addressed urgently. Known also as distributed ledger technology, blockchain technology is characterised by its decentralisation, openness, and transparency, which allows everyone to participate in database records [97]. In 2016, Google proposed a new AI technology called federated learning. During the big data exchange, it can carry out efficient ML among multiple participants or computing nodes to ensure information security and protect the privacy of terminal information and personal information [98]. Decentralised platforms provide important solutions to protect the privacy of patient data in the future.

It is also urgent to standardise deep learning research. In deep learning research, there tends to be a process-based approach, with rules for data collection, the training of models, and the evaluation of the results. Unfortunately, many studies have different criteria for including and excluding original data and related indicators for evaluating their effects. The selection of relevant training models and the direction of network reconstruction remains a subjective judgment. A major reason for this is that there are no standardised guidelines for the research on medical AI. We can build more secure and effective AI systems and realise realistic transformations and applications of these promising technologies only by establishing standardised deep learning process guidelines and innovation.

## 8. Conclusions

The articles we reviewed preliminarily demonstrated the potential of AI methods in diagnosis, treatment, and prognosis of OM. Medical history information, endoscopic otoscopy, radiology, audiology, laboratory data, genetic data, and surgical videos have been used to create AI systems to effectively diagnose patients, aid treatment, and predict treatment response and disease activity. As a result, AI provides an advantage to the quality of care that OM patients currently receive, resulting in improved quality of life and improved OM-related health outcomes. However, most of these studies have been conducted in the laboratory. Therefore, further validation of these results in more realistic Settings and on a larger scale is needed to assess their effectiveness in the general OM population.

## Figures and Tables

**Figure 1 diagnostics-13-02309-f001:**
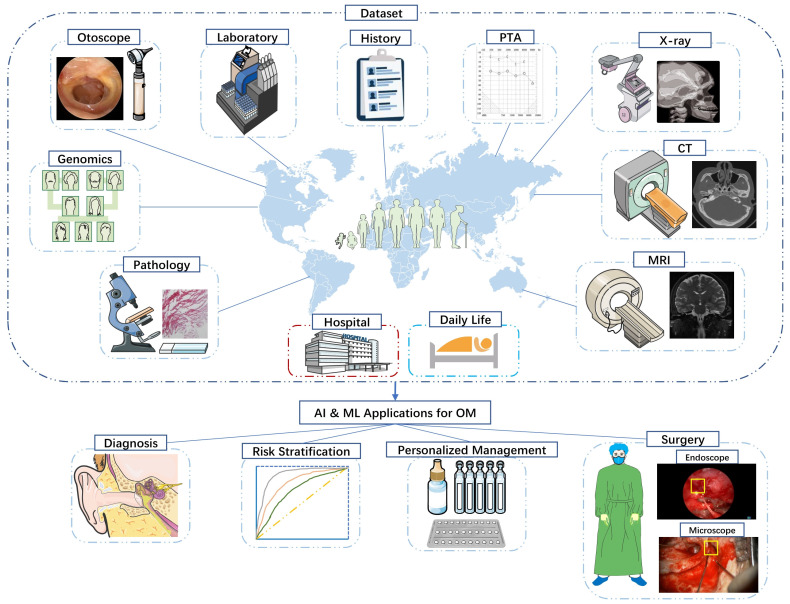
Flow chart of application of AI in diagnosis, treatment and healthcare of OM. Through collecting large amounts of data from hospitals and daily life and training with AI and ML, it can be realized the diagnosis, risk stratification, disease management and treatment of OM.

**Figure 2 diagnostics-13-02309-f002:**
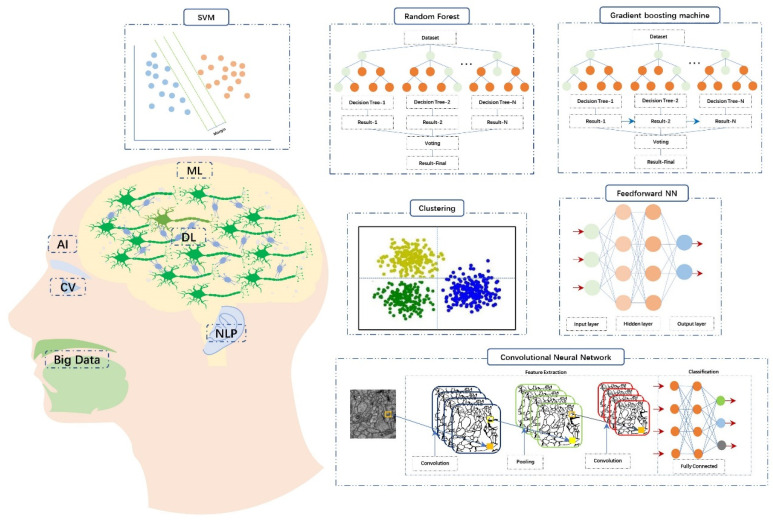
Concepts of artificial intelligence (AI), machine learning (ML), and deep learning (DL). AI is equivalent to the human body, ML equals to the human brain, and DL corresponds to the complex neural circuit consisted of neurons. When AI is applied to vision, it refers to computer vision (CV). And when applied to speech processing, it represents natural language processing (NLP).

**Figure 3 diagnostics-13-02309-f003:**
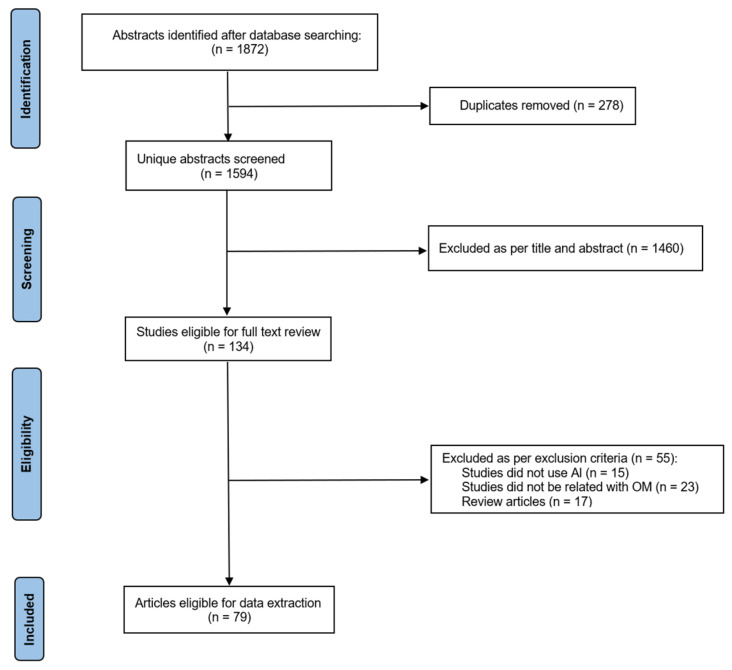
PRISMA flow diagram depicting the different phases of the review selection process.

**Table 1 diagnostics-13-02309-t001:** Types of machine learning algorithms.

ML Algorithms	Definition
**Supervised learning**	**The model is trained based on the labeled data to make the prediction. It is mainly used for classification (continuous data) and regression problems (discrete data).**
**Ensemble learning**	Generate multiple models and then combine these models according to a certain method. It is mainly applied to improve model performance or reduce the possibility of improper model selection.
***Bagging***-Random forest (RF)	As a type of ensembled learning, it is based on decision trees. Each of decision trees can produce an independent and de-correlated output. And then the final result is determined by the rule of “majority-vote”. Each decision tree has the equal weight.
***Boosting***-Gradient boosting machine (GBM)	The basic unit is the decision tree. The performance of the first decision tree is only due to random guessing. The decision tree model established later are based on the prior one by adjusting parameters to gradually improve the accuracy of subsequent models. And then the final result is determined by considering comprehensively the predicted values of all decision trees. Base learner (decision tree) with better performance has higher weight.
Support vector machine (SVM)	In a given set of training samples, with functional distance as constraint condition and geometric distance as objective function, the maximum-margin hyperplane with the best generalization ability and the strongest robust is determined, so as to realize the binary classification of data.
**Unsupervised learning**	**The training data is not labeled and the goal is to find patterns in the data. It is mainly used to solve clustering and dimension reduction problems.**
Clustering	The process of sorting data into different classes or clusters based on mathematical relevance.
Dimensionality reduction	It means converting high-dimensional data into a lower representation with fewer features and is often used for data visualization or data preprocessing.
**Reinforcement learning**	**Through interaction with its environment and trial and error, the training model learns the environment-to-action reflex that maximize cumulative returns.**
**Transfer learning (TL)**	**Transfer the learned knowledge from a domain to another and save a deal of time and computing resources for the training.**
**Deep learning (DL)**	**Mimicking the hierarchical network of neurons in the brain and using multiple layers of data processing, the model can automatically detect the required features and predict the result. But it requires larger quantities of data and advanced computational capacity.**
Feedforward neural networks (FNN)	Arranged in layers, each neuron receives only the output of the previous layer and sends it to the next layer. There is no feedback between adjacent layers.
Convolutional neural networks (CNN)	It is often used in imaging analysis. Through the convolution layer, activation function and pooling layer, the input image is approximate processed and redundant features are removed to reduce the computational complexity and help prevent overfitting. And then the significant features are combined through the fully connected layer to output prediction or classification.
**Natural language processing (NLP)**	**Algorithms are established to organize and interpret human language. The aim is to realize the interactive communication between humans and machines.**
**Computer vision (CV)**	**Algorithms are established to enable the computer to perceive, observe and understand the environment through images and vision, and finally have the ability to adapt to the environment.**

## Data Availability

Not applicable.

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
