# Peer review of "Diagnosis, Treatment, and Management of Otitis Media with Artificial Intelligence"

_diagnostics, 2023, doi:10.3390/diagnostics13132309_

Round 1
Reviewer 1 Report
This is a scoping review of diagnosis, treatment, and management of otitis media with artificial intelligence.
The abstract is adequate in length and structure.
The introduction section addresses appropriate issues, is succint and ends with aims of the paper highlighted adequately.
The Materials and Methods section details the search strategy used, but fails to mention PRISMA guidelines. A flowchart should also accompany literature results.
The discussion has embedded a lot of cited research and is plagued by over-reaching conclusions. However, since this is a review paper aiming at condensing a state-of-the-art review of current AI diagnostic and treatment modalities, such heterogenous sections on various themes are not unacceptable.
Reviewer 2 Report
The following points need to be addressed in the revision:
- The abstract needs revision elaborating the extensively summarized sentences. A refined version must also include at least the names of the algorithms most commonly employed in OM.
- The motivation is weak in the abstract, although mentioned in the introduction. The treatment is carried out with well-known medicines. So what are the hazards if OM is not diagnosed or treated earlier?
- Many typos and grammatical mistakes in the article need to be corrected.
- Add the following reference at the end of Line 32 in support of the idea for intraoperative surgery.
https://doi.org/10.3390/app12083715
- Rephrase the heading in sections 3.1, 3.2, and 3.3 making them more elaborative and useful for the readers.
- A nomenclature table needs to be added describing the short names used in the article.
- Line 74, why not write “4.1 Computer Vision” instead of “4.1 CV-based”
- Tables s1 to s6 are absent. Without these tables, the article cannot be accessed for its publication.
- Add some literature references in section 3.2.
- A separate section for datasets publicly available must be provided with references for OM.
- A comparison of existing SOTA techniques should be provided at the end before Conclusions showing the results of different techniques.
- In future recommendations, add references like XAI, Radio-genomics datasets for checking the chemo-resistance in glioblastoma patients, and Fluorescent nanodiamonds used with AI for the treatment of multiple problem domains. These possible trends may be suggested for OM treatment as future horizons.
- Conclusions must be added at the end of the article.
Minor revision in the language is suggested.
Round 2
Reviewer 2 Report
The paper is in shape now, and a real contribution to the field of otitis.
Some minor changes will further improve the article:
1. Replace “. And several ” with “Several ” in Line 14 (page 1) Abstract
2. Line 98: Add one or two lines about what the readers would go through in this entire Main Section 4
3. Line 99: Add a little detail in just one or two lines, very briefly, about the mini sections to come ahead. This would attract the audience to further go through the article.
4. Replace “4.1.1. otoscopy” with “4.1.1 Otoscopy” in Line 100
5. Line 136: Make the heading “4.1.2. CT、MRI、Pathology” more descriptive
